# Salvage Whole-Pelvic Radiation and Long-Term Androgen-Deprivation Therapy in the Management of High-Risk Prostate Cancer: Long-Term Update of the McGill 0913 Study

Sara Elakshar [1,2], Marwan Tolba [1] , Steven Tisseverasinghe [3], Laurie Pruneau [1] , Vanessa Di Lalla [1], Boris Bahoric [1] and Tamim Niazi [1,*]

1   Department of Radiation Oncology, Jewish General Hospital, McGill University, Montreal, QC H3T 1E2, Canada; sara.elakshar@med.tanta.edu.eg (S.E.); marwana.tolba@nshealth.ca (M.T.); laurie.pruneau@umontreal.ca (L.P.); vanessa.dilalla@mail.mcgill.ca (V.D.L.); boris.bahoric.med@ssss.gouv.qc.ca (B.B.)

2   Department of Clinical Oncology, Tanta University Hospitals, Tanta University, Tanta 6632110, Egypt

3   Department of Radiation Oncology, Gatineau Hospital, McGill University, Gatineau, QC J8P 7H2, Canada; steven.tisseverasinghe@ssss.gouv.qc.ca

*   Correspondence: mohammad.tamim.niazi.med@ssss.gouv.qc.ca; Tel.: +1-514-340-8222; Fax: +1-514-340-7548

**Abstract:** Purpose: To report the long-term outcomes of the McGill 0913 study and the potential benefits of combining prostate-bed radiotherapy (PBRT), pelvic-lymph-node radiotherapy (PLNRT), and long term ADT (LT-ADT). Materials and Methods: From 2010 to 2016, 46 high-risk prostate cancer patients who experienced biochemical recurrence (BCR) after radical prostatectomy (RP) were enrolled in this single-arm phase II clinical trial. The patients were eligible if they had a Gleason score > 8, locally advanced disease (≥pT3), a preoperative PSA of >20 ng/mL, or positive lymph nodes (LN). The patients were treated with a combination of 24 months of ADT, PBRT, and PLNRT. The primary outcome was biochemical progression-free survival (bPFS) and the predefined secondary endpoints included distant-metastasis-free survival (DMFS), overall survival (OS), and toxicity. In this update, we also report the median follow-up of 8.8 years and 10 years OS. Results: At a median follow-up of 8.8 years, 43 patients were eligible for analysis. The median pre-salvage PSA was 0.30 μg/L. Half (51%) of the patients (*n* = 22) had positive margins, 40% (*n* = 17) had Gleason scores > 8, 63% (*n* = 27) had extracapsular extension, 42% (*n* = 18) had seminal vesicle invasion, and 19% (*n* = 8) had LN involvement. The 10-year bPFS was 68.3 %. The 10-year DMFS was 72.9%. The 10-year OS was 97%. There were two non-cancer-related deaths. The first patient died of congestive heart failure while the other died of an unknown cause. No new toxicity was observed after the initial report. Conclusions: Our study demonstrates that treatment escalation with PBRT, PLNRT, and LT-ADT improves long term outcomes. In view of the recently published SPPORT study, we conclude that this novel approach of treatment intensification in high-risk post-prostatectomy patients is safe and effective, and that it should be offered as the standard of care.

**Keywords:** prostate; cancer; radiotherapy; hormonal therapy; salvage; radical prostatectomy

## 1. Introduction

Prostate cancer remains the most common non-cutaneous malignancy diagnosed amongst the male population [1]. The treatment of prostate cancer encompasses several reliable treatment options for patients afflicted with different disease stages, accounting for and optimized to different patient characteristics, such as performance status, risk group, and clinical and radiological stage, as well as individual patient preferences.

Radical prostatectomy (RP) is a common and effective definitive treatment, offered to suitable patients after appropriate selection. While it is an excellent option for early-stage

disease, early signs of treatment failure can be detected through a rise in PSA meeting the threshold of biochemical failure. However, newer data have shown that these failures can be effectively managed by a combination of salvage external beam radiotherapy (RT) and/or androgen-deprivation therapy (ADT).

Over the past decade, there has been a remarkable evolution in imaging techniques aiming to more accurately stage prostate cancer patients. Innovative imaging technologies, such as positron emission tomography (PET) and PET/computed tomography (CT), have shown promising improvements in both sensitivity and specificity when compared to traditional imaging.

Preliminary research suggests that PET tracers designed specifically for prostate cancer, such as F-18 fluciclovine, C-11 choline, and Ga-68 PSMA-11, may prove to be superior to FDG in cases of recurrent prostate cancer [2–6].

The F-18 fluciclovine (Axumin), an amino acid analog, has been shown to exhibit enhanced uptake by cancer cells. In a study involving 143 prostate cancer patients with PSA-only recurrence, F-18 fluciclovine demonstrated a sensitivity of 91% and a specificity of 40% [7].

Notably, the EMPIRE-1 trial demonstrated that incorporating F-18 fluciclovine PET imaging into radiotherapy-decision making and planning for men with increasing PSA levels after prostatectomy significantly improved outcomes from salvage RT in patients without evidence of extrapelvic disease via conventional imaging [8].

Moreover, the PET tracers, F-18 and C-11 choline, which target cell-membrane-lipid biosynthesis, which is known to be increased in cancer cells, have shown promise. A systematic review of 47 studies involving 3167 patients revealed that C-11 choline or F-18 choline PET/CT led to a modification of the treatment plan in 41% of patients [9].

Advancements in PET scanning using novel radiotracers targeting PSMA, such as Ga-68 PSMA-11 (gozetotide) and F-18 DCFPyL (piflufolastat F-18), have shown potential in detecting both locoregional and distant metastatic sites, even in prostate cancer patients with very low levels of PSA (<2 ng/mL). A systematic review of 15 trials involving 1163 prostate cancer patients revealed that Ga-68 PSMA-11 PET played a crucial role in managing patients with biochemical failure following initial local therapy, with changes in management observed in 54% of the cases [3].

Indeed, after proper restaging, salvage RT has the potential to offer durable disease control in cases of recurrent prostate cancer, provided that the recurrence is confined within the treatment field and that an adequate radiation dose is administered to eliminate the residual cancer [10].

The combination of prostate-bed radiotherapy (PBRT) with androgen-deprivation therapy has been investigated in several large, randomized phase III trials. The GETUG-AFU 16 trial showed a benefit in DMFS when short-term ADT (ST-ADT) was added to locoregional treatment (PBRT with pelvic-lymph-node radiotherapy (PLNRT) or previous LN dissection). Indeed, the study demonstrated a progression-free survival (PFS) rate of 64% at 10 years for patients treated with RT and ST-ADT, whereas the PFS was only 49% in those treated with RT alone. The difference was statistically significant with a *p*-value of <0.0001 [11]. The RTOG 0534 (SPPORT trial), which enrolled almost 1800 patients, randomly assigned its participants to three treatment groups: PBRT alone, PBRT + ST-ADT, and PBRT + PLNRT + ST-ADT. The authors of the SPPORT trial noted a biochemical-progression-free survival (bPFS) advantage when adding PLNRT to PBRT and ST-ADT, compared to PBRT and ST-ADT alone [12]. The long-term results of the RTOG 9601, showed a 5% overall survival benefit when the patients received long-term ADT (LT-ADT) combined with daily bicalutamide for 24 months with salvage PBRT. The RTOG 9601 was the first major trial to highlight the potential long-term benefits of combining these treatment modalities [13]. Collectively, these studies provide significant evidence for the effectiveness of the combined use of PBRT and ADT, emphasizing the need for further exploration of this therapeutic approach in patients with prostate cancer. In the McGill 0913 study, treatment was intensified with both LT-ADT and PLNRT for prostate cancer

patients at high risk of relapse. We hypothesized that a longer duration of ADT and PLNRT may improve outcomes and, potentially, overall survival [13]. In this analysis, we report the long-term results and updated survival outcomes of the McGill 0913 trial.

## 2. Materials and Methods

### 2.1. Patient Population

Between 2010 and 2016, 46 high-risk prostate cancer patients who had BCR after RP approved to participate in the McGill 0913 study. Patients were eligible if they had an initial PSA $\geq$ 20 µg/L, Gleason score $\geq$ 8, and/or a pathological T-stage $\geq$ T3. Patients needed to have adequate functional status (Karnofsky performance score (KPS) > 70) and marrow function (platelets $\geq$ 100,000 cells/mm$^3$, hemoglobin $\geq$ 10.0 g/dL, and AST/ALT < 2 times the upper limit of normal). Patients with pelvic lymphadenopathy $\geq$ 1.5 cm, distant metastasis on imaging, detectable residual tumor, post-operative PSA $\geq$ 5.0 µg/L, prior pelvic radiotherapy, other malignancies within the past 5 years, or active severe comorbidities were excluded.

### 2.2. Treatment

Patients started Bicalutamide 50 mg within 2 weeks of enrollment. After a total of 2–4 weeks of Bicalutamide, patients received a luteinizing hormone-releasing hormone (LHRH)-agonist injection, which was then administered every 3 months for 24 months. External beam radiotherapy (EBRT) was started 8–12 weeks after the first LHRH agonist injection. The EBRT was delivered in two phases. Fifteen patients were treated by using 3D conformal radiotherapy, while the rest of the patients were treated via newer techniques, either IMRT or VMAT. The pelvis was treated with 44 Gy in 22 fractions in the first phase. The pelvic clinical target volume (CTV) encompassed the prostate bed, the remnants of the seminal vesicle, and the pelvic lymph nodes (LNs), which were contoured as per the RTOG pelvic LN guideline [14]. In the second phase, the prostate bed only was boosted to 22 Gy in 11 fractions. Radiation was delivered using $\geq$6 MV photons. Prior to 2012, a 3D conformal radiotherapy (3DCRT) technique was used, which was later replaced with an intensity-modulated radiotherapy (IMRT) technique. The dose was prescribed to the isocenter for 3DCRT plans and to the planning target volume (PTV) for the IMRT plans (i.e., 100% of the prescribed dose to cover 95% of PTV).

### 2.3. Assessments

Pre-treatment assessments included a detailed history, physical exam, and completion of QOL questionnaires, such as the EORTC QLQ-C30 (11), the EQ-5D (12), the International Index of Erectile Function (IIEF-5) [15], and the International Prostate Symptom Score (IPSS)] [16]. Investigations included serological assessments (CBC, electrolytes, urea, liver-function tests, testosterone levels, and PSA), chest x-ray, CT or MRI of the abdomen and pelvis, and a whole-body bone scan. During radiotherapy, patients were seen weekly by their radiation oncologist. Observed toxicities were reported using the Common Toxicity Criteria version 3.0 scale (CTCAE v3.0) [17]. Patients were assessed at post-treatment follow-up visits every 3 months for a period of 2 years, every 6 months for an additional 3 years, and annually thereafter. During each follow-up visit, a range of assessments were performed, including PSA, blood testosterone levels, EQ-5D, IPSS, IIEF-5, and toxicity. Furthermore, the EORTC QLQ-C30 questionnaire was administered and completed on an annual basis.

### 2.4. Endpoints and Statistical Analysis

The primary objective of the McGill 0913 study was bPFS. Secondary outcomes included DMFS, OS, QOL, and toxicity. The bPFS was calculated from the date of the first injection of LHRH agonist to the date of biochemical recurrence. The DMFS and OS were calculated from the date of the first LHRH-agonist injection to the date of radiological progression or death from any cause, respectively. The Kaplan–Meier method using IBM SPSS v24 was used for the outcome analysis.

## 3. Results

Between 2010 and 2016, forty-six patients participated in this study. Forty-three patients are included in the current analysis. Three patients were ineligible in the initial analysis, since they were not treated according to the protocol. Of the three, one opted for salvage surgery, while the other two refused radiotherapy. Baseline characteristics were evaluated for all the patients in the analysis (Table 1).

**Table 1.** Participants' baseline clinical and pathological features.

| Characteristic | *n* = 43 |
|---|---|
| Age | |
|   Median (IQR) | 65 (59–69) |
| Pathological T stage, n (%) | |
|   T1 | 2 (4.7%) |
|   T2 | 11 (25.6%) |
|   T3 | 30 (69.8%) |
| Biopsy Gleason score | |
|   7 | 26 (60.5%) |
|   8 | 12 (27.9%) |
|   9 | 5 (11.6%) |
| Pathological Gleason score | |
|   6 | 1 (2.3%) |
|   7 | 25 (58.1%) |
|   8 | 9 (20.9%) |
|   9 | 6 (14%) |
|   Unknown | 2 (4.7%) |
| Testosterone (nM/L) | |
|   Median (IQR) | 11.2 (8.28–16.10) |
|   Mean (SD) | 12.31 (5.25) |
|   Unknown | 2 (4.7%) |
| Postoperative PSA | |
|   Median (IQR) | 0.3 (0.2–0.47) |
| Margin status | |
|   Negative | 18 (41.9%) |
|   Positive | 22 (51.2%) |
|   Unknown | 3 (7%) |
| LN involvement | |
|   No | 29 (67.4%) |
|   Yes | 8 (18.6%) |
|   Unknown | 6 (14%) |
| ECE | |
|   No | 9 (20.9%) |
|   Yes | 27 (62.8%) |
|   Unknown | 7 (16.3%) |
| SV involvement | |
|   No | 23 (53.5%) |
|   Yes | 18 (41.9%) |
|   Unknown | 2 (4.7%) |
| Time from surgery to post-operative therapy | |
|   Median weeks (IQR) | 68.3 (27–177.6) |
| Duration of ADT prior to starting RT | |
|   Median weeks (IQR) | 9 (8–10.9) |

IQR: interquartile range, SD: standard deviation, PSA: prostate-specific antigen, LN: lymph nodes, ECE: extracapsular extension, SV: seminal vesicle, ADT: androgen-deprivation therapy.

The high-risk pathological features present in the patient population included positive lymph nodes in 19% (*n* = 8), Gleason scores ≥ 8 in 35% (*n* = 15), extracapsular extension in 63% (*n* = 27), and seminal vesicle involvement in 42% (*n* = 18) (Table 1). The postoperative median PSA was 0.30 µg/L (interquartile range (IQR) −0.20 to 0.47).

At a median follow-up of 8.8 years (3.25–12 years), the 10-year bPFS, DMFS, and OS were 68.3%, 72.9%, and 97%, respectively (Figures 1–3).

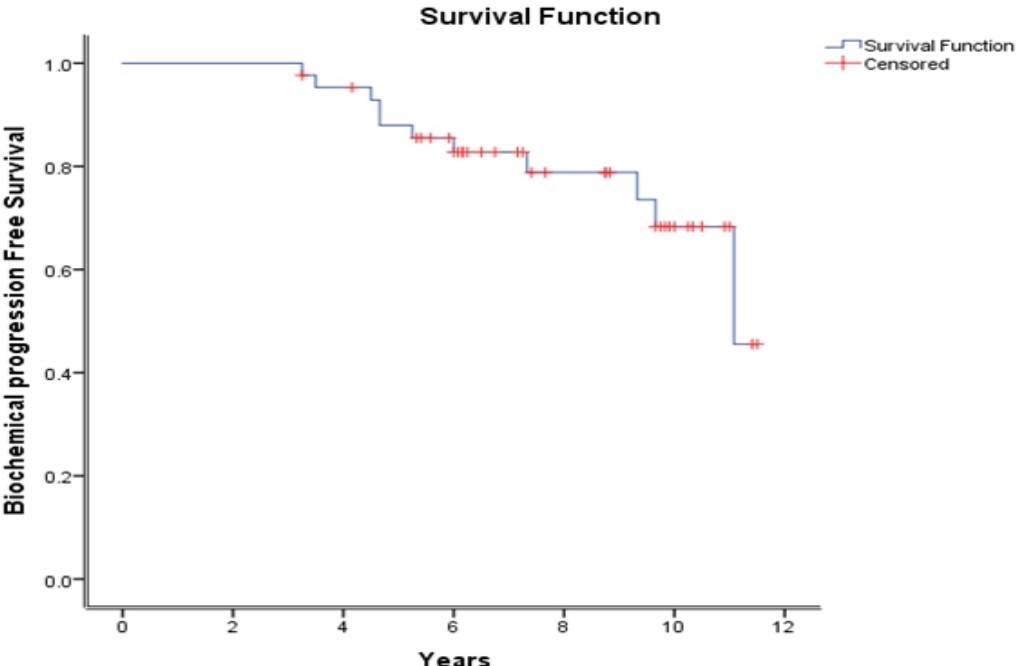

**Figure 1.** Biochemical progression-free survival over time as per the Phoenix definition (prostate-specific-antigen nadir + 0.2 µg/L).

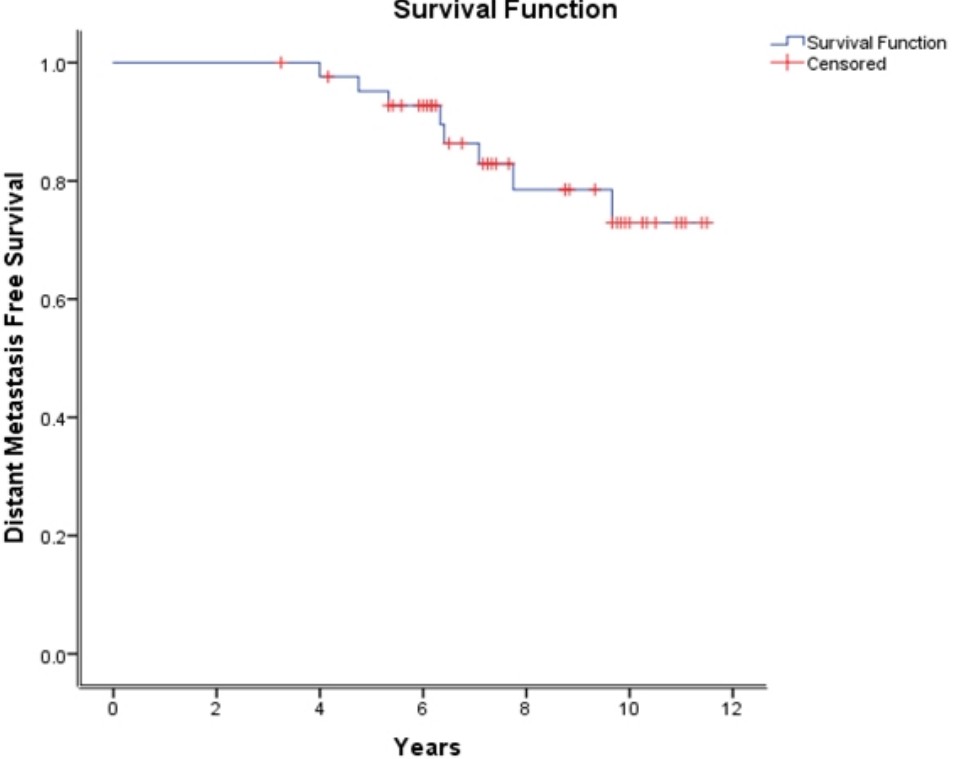

**Figure 2.** Distant metastasis-free survival over time, defined radiologically.

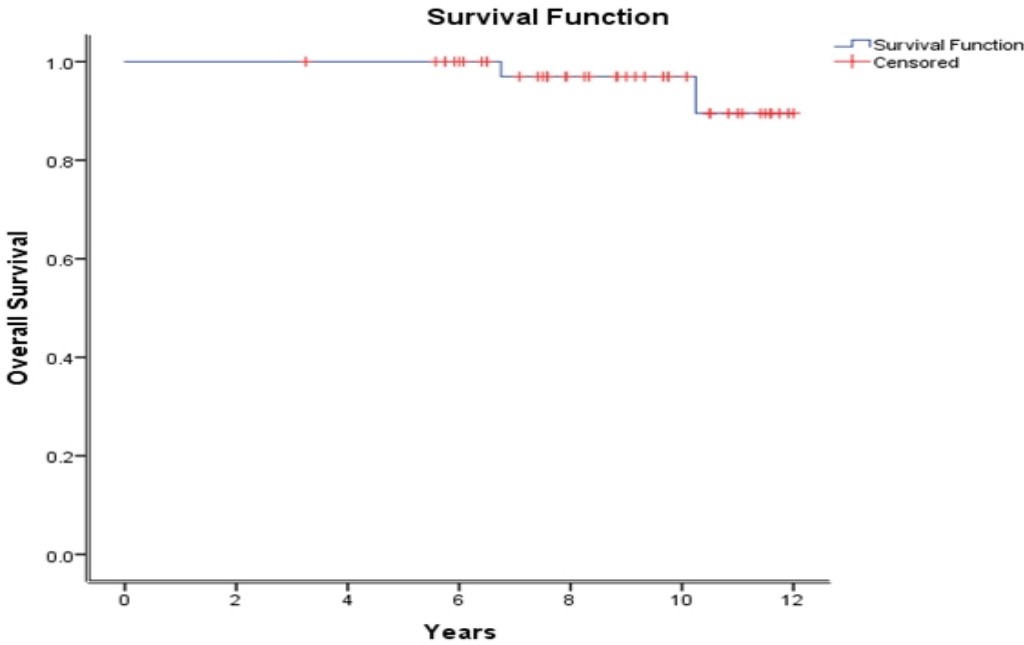

**Figure 3.** Overall survival from first injection of LHRH agonist until death or last follow-up.

No local or pelvic relapses were detected in this study. In fact, the distant failures involved either non-regional LNs or the bones. All the patients who developed bone metastases had Gleason scores ≥ 8 (Table 2). While two patients died of other causes, no patients died from prostate cancer. The first patient died of congestive heart failure at the age of 83, with 10.17 years of follow-up. The second died of an unknown cause at the age of 80, with 6.75 years of follow-up.

**Table 2.** Descriptive details of distant-metastases events.

| Patient No. | Gleason Score | pT-Stage | Nodal Status | Margin Positivity | iPSA | Site of Metastases | Time of Event (Months) |
|---|---|---|---|---|---|---|---|
| 5 | 8 (4 + 4) | T3b | N/A | Negative | 5.7 | Hilar LN | 76 |
| 10 | 8 (4 + 4) | T3a | Negative | Negative | 9 | PALN | 85 |
| 19 | 7 (4 + 3) | T3a | Negative | Positive | 7 | PALN | 116 |
| 22 | 8 (4 + 4) | T3b | Negative | Positive | 5 | Scapula | 93 |
| 23 | 7 (3 + 4) | T3b | N/A | Positive | 5.7 | PALN | 64 |
| 33 | 8 (4 + 4) | T3a | Negative | Negative | 6 | T11 | 77 |
| 41 | 9 (4 + 5) | T3a | Negative | Negative | 9.3 | 12th Right rib | 57 |
| 42 | 8 (4 + 4) | T3a | N/A | Negative | 5.2 | Left pubic ramus | 48 |

iPSA: initial prostate-specific antigen; pT stage: pathological T-stage; PALN: para-aortic lymph node.

Since the prior report, no new toxicity events were observed. The rates of Grade 2 ADT-related or radiation-induced toxicities are reported in Table 3. Overall, the long-term toxicity profiles of the enrollees was acceptable. The QOL, assessed by the EQ-5D's visual analog score [18], remained unchanged. While the ADT seemed to be associated with a decrease in QOL, this was not statistically significant from the baseline ($p$ = 0.39) (Figure 4). The mean minimum QOL experienced by a patient at any given time while on ADT was 7.8 (standard deviation = 2.0), compared to a mean baseline QoL of 8.2 (standard deviation = 1.2).

**Table 3.** Patients which experienced new grade 2 or higher toxicities, related to intervention.

| | During ADT (%) | Post ADT (%) |
|---|---|---|
| Grade 2 or higher ADT-induced toxicity (Hot flushes, ED and/or fatigue) | 5 (10.8%) | 1 (2.2%) |
| | Acute (<90 days) | Late (>90 days) |
| Grade 2 or greater radiation-induced toxicity | 2 (4.3 %) | 2 (4.3 %) |

ADT: androgen-deprivation therapy, ED: erectile dysfunction.

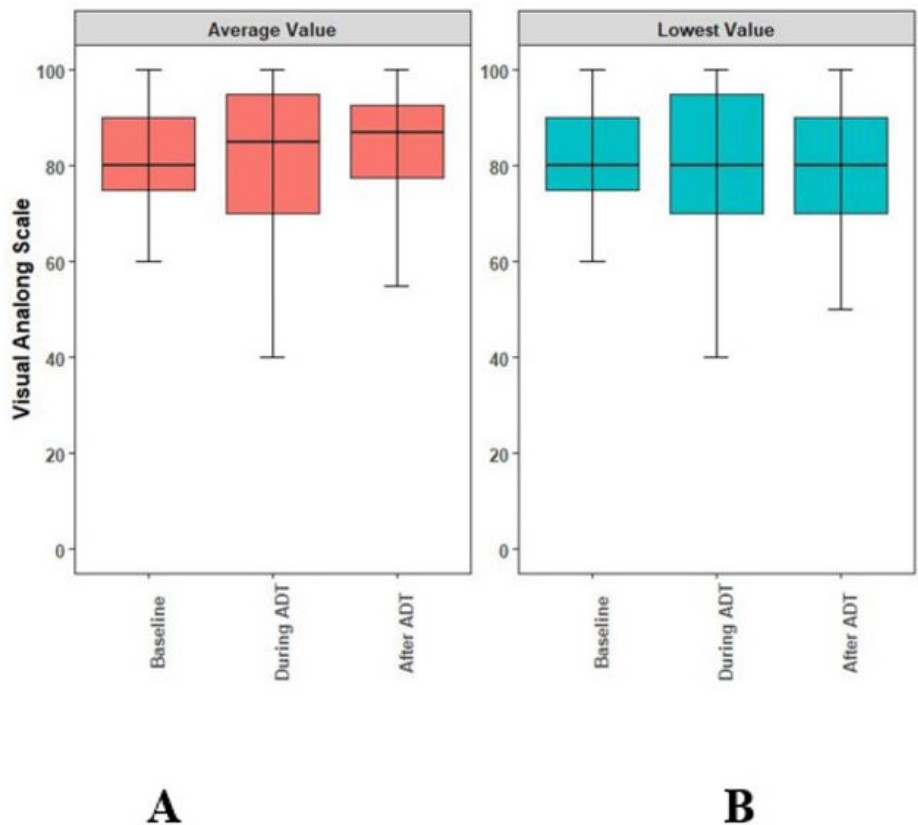

**Figure 4.** Boxplots illustrating the medians, interquartile ranges, and ranges of quality of life reported by individual study participants at baseline, during ADT, and any time following ADT. (**A**) Comparison of the averaged self-reported QoL, (**B**) comparison of the lowest self-reported QoL. ADT: androgen-deprivation therapy. QoL: quality of life.

## 4. Discussion

The benefit of combining PLNRT with local prostate RT and ADT in both salvage and definitive settings has long been debated. The use of LT-ADT has a proven survival benefit compared with ST-ADT in patients with high-risk prostate cancer receiving definitive RT to the prostate and pelvic lymph nodes [19]. In a recently published phase III randomized controlled trial (POP- RT), PLNRT with LT-ADT in high-risk and very high-risk prostate cancer demonstrated a 14% benefit in biochemical failure-free survival (BFFS) and a 7% benefit in disease-free survival (DFS) at 5 years [20]. Furthermore, the benefit of PLNRT in pathological or clinical node-positive disease was also suggested in a subgroup analysis of STAMPEDE [21]. This partly contrasted with prior negative trials (GETUG-01 and RTOG 9413), which used ST-ADT combined with PLNRT. In fact, GETUG-01 used 4–8 months of ADT [22], while RTOG 9413 used 4 months of ADT [23].

Regarding the effectiveness of PLNRT, the ongoing NRG0924 study is an attempt to provide a definitive answer. The NRG0924 trial aims to accrue 2580 patients to demon-

strate whether prophylactic neoadjuvant ADT and PLNRT can improve OS in unfavorable intermediate- or high-risk disease.

While numerous trials have investigated different combinations of RT and ADT in the salvage setting, the McGill 0913 was the first to prospectively evaluate PBRT with both PLNRT and LT-ADT. Compared to the aforementioned mentioned data, this treatment combination demonstrated encouraging disease control without significantly affecting patients' QoL. Our results add further credence to treatment intensification for post-prostatectomy patients at high risk of relapse [24].

In the RTOG 9601 trial, the cumulative incidence of biochemical recurrence after salvage RT at 12 years was 44.0% in the bicalutamide group. When comparing arms, the 5-year PSA failure improved from ~50% to 23% (hazard ratio (HR) = 0.48; 95% CI 0.40–0.58).

Our 10-year bPFS rate was 68.3 %, which compares favorably with that of the RTOG 9601 trial at 10 years. While direct comparisons between trials are not valid, differences may be attributed to PBRT only without PLNRT [13].

The DMFS has recently emerged as a robust surrogate for OS in prostate cancer [20,25,26]. In the GETUG-AFU 16 trial, the 10-year metastasis-free survival was 75% (95% CI 70–80). In the arm comprising radiotherapy plus 6 months of Goserelin, an additional benefit in DMFS of 6% was found [11]. This is comparable to our DMFS rate of 72.9% at 10 years. In the GETUF-AFU 16 study, the patients who did not undergo a pelvic nodal dissection during their radical prostatectomy and whose risk of nodal involvement was over 15% on the Partin table then received pelvic nodal radiotherapy [27].

In the RTOG 9601, the OS actuarial rate at 12 years was 76.3% in the bicalutamide group [13]. In parallel, our 10-year OS rate remains excellent at 97%. Similarly, our 10-year OS rate compares favorably with that of the GETUG-AFU 16 trial. The patients assigned radiotherapy plus Goserelin had a 10-year OS rate of 86% [11].

Regarding the treatment of node-positive patients after radical prostatectomy, 36 institutes in the USA took part in a study to assess whether immediate ADT offers benefits in terms of OS compared with deferred ADT. After a median follow-up of almost 12 years, early ADT led to significant improvements in OS compared to deferred ADT in patients with nodal metastases after radical prostatectomy.

According to the latest published NCCN prostate cancer guidelines, CT, MRI, PET/CT, or PET/MRI with F-18 NaF C-11 choline, F-18 fluciclovine, Ga-68 PSMA-11, or F-18 piflufolastat PSMA can be considered for evaluating equivocal results on initial bone imaging [4]. For the soft-tissue imaging of the chest, abdomen, and pelvis, CT chest and abdominal/pelvic MRI are preferred. Alternatively, Ga-68 PSMA-11, F-18 piflufolastat PSMA PET/CT, or PET/MRI can be considered for bone and soft-tissue imaging.

Due to the higher sensitivity and specificity of PSMA PET tracers in detecting micrometastatic disease compared to conventional imaging (CT, MRI) during initial staging, conventional imaging is not an essential prerequisite for PSMA PET. Therefore, PSMA PET can now be considered an equally effective, if not more effective, imaging tool.

The RADICALS-HD study was part of the RADICALS protocol. The RADICALS design contained two separate randomizations seeking to answer different questions on radiation timing and ADT duration in the post-operative setting in a similar patient population. The RADICALS study overall examined nearly 4000 patients with a follow-up of fifteen years. The first randomization assessed the benefit of adjuvant versus salvage radiotherapy, thereby comparing the differences in the timing of the initiation of the postoperative RT. This question was evaluated through the RADICALS-RT study. The second randomization sought to address the optimal duration of hormone therapy through three arms, using either 0, 6, or 24 months of ADT. An overview of this question was presented in the report on the RADICALS -HD trial. The RADICALS HD study assessed 2839 patients from the UK, Canada, and Denmark. Of these, 23% had high-risk features, including pT3b or T4 disease whereas, 20% had Gleason 8–10 disease. The median age of the enrolled patients was 66 years, whereas the median PSA prior to salvage was 0.22 ng/mL [28]. The patients who were eligible to receive postoperative RT were randomized to either

NO ADT, ST-ADT, or LT-ADT. When comparing the ST-ADT arm to the LT-ADT arm, 2 years of ADT improved MFS. Indeed, 24 months of ADT improved MFS, with a HR of 0.77, and a 95% confidence interval (CI) = 0.61–0.97, with $p = 0.3$. When comparing the no ADT and the ST-ADT groups, no benefit was seen in terms of MFS, with a HR of 0.89 and a CI: 0.69–1.14. This corresponded to 10-year event-free survival rates of 79% vs. 80%, respectively. Furthermore, the MFS at 10 years was 78% in the 6-month group vs. 72% in the 24-month group. The time-to-salvage ADT was delayed in both comparisons, with a HR of 0.73 and a CI 0.59–0.91. No OS benefits were noted in the groups when adding either ST-ADT or LT-ADT, with a HR 0.88 and a CI: 0.66–1.17 [28]. Interestingly, the RADICALs-HD trail permitted randomization through a three-arm comparison between 0, 6, and 24 months of ADT and two subgroups comparing 0 vs. 6 months and 6 vs. 24 months. The three-way randomization regrouped 492 patients, whereas the randomization into the two subgroups regrouped the 2347 patients, representing 83% of the total RADICAL-HD patient population. A further subgroup analysis showed that the patients with high-risk features were more likely to receive 6 to 24 months of ADT rather than 0 to 6 months of ADT, probably due to inherent physician bias. Indeed, the patients with Gleason 8–10 disease represented 11% vs. 28% of the cohorts in the ADT-duration arms of 0– 6 vs. 6– 24 months. The disease stages of T3b or T4 characterized 16% vs. 30% of the patients in the ADT-duration arms of 0–6 vs. 6– 24 months. Similarly, 3% vs. 8% of the patients in the ADT-duration cohorts of 0–6 vs. 6– 24 months had positive lymph nodes as part of their pathological characteristics. Hence, while the results of the RADICALs-HD trial are promising, 24 months of ADT should not necessarily be applied to all patients requiring salvage treatments, but it should be considered preferable in higher-risk patients [28].

Our results are further supported by the recent final publication of the SPPORT trial [29]. In this study, node-negative T2–T3 prostate cancer patients with a PSA of 0.1–2.0 µg/L (median 0.34 µg/L) were randomized to either PBRT alone, PBRT with ST-ADT, or PBRT with both ST-ADT and PLNRT [12]. At a median follow-up of 8.2 years, the 5-year freedom-from-progression rates were 70.9% with PBRT alone, 81.3% with PBRT and ST-ADT, and 87.4% with PBRT, PLNRT, and ST-ADT. Like our results, the addition of PLNRT to PBRT and ST-ADT was found to further improve freedom from progression [29]. Our trial distinguished itself from other publications by including positive-lymph-node prostate cancer patients. The results of our trial indicate that treatment intensification showed equally good outcomes for node-positive postoperative patients with intrinsically higher-risk disease. The SPPORT [20] and ROTG 9601 [5] included only node-negative patients and nearly 20% of the patients in GETUG-AFU16 [3], pN0 or pNx, and McGill 0913 were lymph-node-positive. However, our results are comparable to these large, randomized trials.

In our final analysis, we did not find any evidence of newly emerging toxicity. There were also no statistically significant changes in the patients' overall QoL assessments compared to their baseline values [24]. Our study has a few limitations, some of which are inherent to its phase II design. Indeed, the study had a small sample size, with only 46 participants. The study was open-label, with a single experimental arm and no comparator. Therefore, an external standard is required for a comparison with our results, and our trials main hypothesis-generating. This study was limited to a single academic institution, which may limit its external validity. However, our results are both promising and intriguing, particularly given our excellent results for DMFS and OS, despite the inclusion of a substantial portion of node-positive patients.

## 5. Conclusions

The results of McGill 0913 are comparable to the recently published SPPORT randomized clinical trial, despite its inclusion of only high-risk and lymph-node-positive prostate cancer patients. In themselves, our results, given the above-mentioned limitations, are not sufficient to suggest a change in the standard salvage-treatment approach. However, by combining our results with RTOG 9601 and recently published SPPORT study, we can

conclude that high-risk prostate cancer patients with BCR post-RP should be treated with pelvic and prostate-bed radiation therapy in conjunction with long-term ADT until the data on systemic-therapy intensification emerge. However, given the small sample size inherent to this phase 2 study, these data remain hypothesis-generating and should be further validated in a phase 3 randomized controlled study.

**Author Contributions:** Conceptualization, S.E. and M.T.; methodology, V.D.L.; software, S.E.; validation, T.N., M.T. and S.T.; formal analysis, S.E.; investigation, S.T.; resources, V.D.L.; data curation, M.T. and L.P.; writing—original draft preparation, S.T. and L.P.; writing—review and editing, T.N. and B.B.; visualization, V.D.L.; supervision, T.N. and B.B.; project administration, T.N. All authors have read and agreed to the published version of the manuscript.

**Funding:** Abbvie pharmaceuticals provided funding for the operating budget of this study. The funder was not involved in the study design, the collection, analysis, and interpretation of data, the writing of this article, or the decision to submit it for publication.

**Institutional Review Board Statement:** The studies involving human participants were reviewed and approved by Research Ethics Board of the Jewish General Hospital, Montreal, Canada. Trial registration NCT01255891.

**Informed Consent Statement:** Informed consent was obtained from all subjects involved in the study.

**Data Availability Statement:** The data presented in this study is available on request from the corresponding author.

**Conflicts of Interest:** The authors declare no conflict of interest.

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
