# Peer review of "Salvage Whole-Pelvic Radiation and Long-Term Androgen-Deprivation Therapy in the Management of High-Risk Prostate Cancer: Long-Term Update of the McGill 0913 Study"

_curroncol, doi:10.3390/curroncol30080526_

Round 1

Reviewer 1 Report

This is a phase II single institution of salvage RT in BR patients from prostatectomy. The generating hypothesis is properly due to there is not consensus on how to treat these subgroup of  patients rightnow.

Some comments and suggestions:

Abstract: ok

Introduction:

- Make some comment on the importance of molecular images, mainly, TC-PSMA in this setting

- line 68. Revised reference 6, because the prior sentence is not adequate. In the last paper published f DART trial there is NOT improvement in OS.

Methods: How many patients were treated with 3DC radiotherapy?

Results:

- Table 1. Complete with more relevant characteristics of initial diagnosis

- revised number (%) of pathological Gleason patients (15 vs 17¡)

- Line 128. Eliminate (Table 2). It is not correlate with prior sentence. Generate confusion

- Table 2. Describe type of ADT toxicities (ej., hot flashes, tied, etc)

Discussion: 

- Even there is a good schematic illustration in Table 2, discuss what recommendations authors will make to diminish distant metastasis (in this study 17% of patients)

- Discuss molecular images (TC-PSMA). The importance to include it at diagnosis and why

- Discuss your study with more relevant retrospective pN1 patients, considering there is nearly 20% included in this phase II. You discussed quite properly with Getug, RTOG 9610, Radicals, Spport, but most of them are pN0

Finally, I considere this phase II very interesting for the scientific community as a generating hypothesis work in this setting

Reviewer 2 Report

The title of the manuscript: „Salvage Pelvic Radiation and Androgen Deprivation Therapy 2 in the Management of High Risk Prostate Cancer: Long-Term 3 Update of the McGill 0913 Study”. It is a well-written, interesting, and important paper with clinically relevant messages about the long-term results of the study above and dealing with the necessity of long-term ADT medication and pelvic nodal RT during salvage treatment of biochemical failure after primary radical prostatectomy. Over the undoubted value of the manuscript the reviewer has several recommendations/notifications. After the corrections I suggest the material to publish because of the real clinical message of it.

The comments/recommendations:

1, Title: I miss some extra definitions from the title, e.g., „whole” pelvic radiation, „long-term” androgen deprivation therapy, „salvage” „biochemical relapse” and „localized” prostate cancer. I feel most important the first two addiction because it will become more warning for the readers.

2, Abstract: -Please sign the role of the preoperative PSA level as eligibility criteria (like in Material and Methods section).

-Line20: 8.8 years median follow-up and (corrected) 10-year OS results. Please refer it in the Material and Methods section as well.

-Line24-25: There were two „no-cancer related” deaths.

–In conclusion the authors stated that the combination „improve long term outcomes”. I think it has rather excellent long-term effect.

3, Introduction: -Line53: I suggest the usage of 10 years rather than 120 months.

4, Material and Methods:  -Line82: What was the duration of bicalutamide treatment?

-Line119: The authors stated that „one opted for surgery”. What kind of surgery (after RP)?

-Table 1.: I missed the preoperative PSA values.

-Line126: Gleason score above 8 in 40%. Please sign that this value is generated from preoperative (biopsy) histology.

-Table2: Please use the same color for the ADT periods and acute and late RT toxicity.

-The very low incidence of toxicity is noticeable. Maybe it would be worthy to discuss it.  

-I feel the Figures too big compared to the tables and the text.

-Table3: „-ve” and „+ve”: these are a little bit strange for the reviewer.

5, Discussion: I suggest some structural reconstruction of the section and/or the usage of subtitles, e.g., the role of pelvic node irradiation, the effect of ADT duration etc. with the summary of present study findings in the end (and not immediately in the second paragraph).

-A feel the over-discussion of the RADICALS trial, meanwhile it is an adjuvant study with not only high-risk patients.

-Line214: “At 10 years follow up, the ST-ADT arm had improved MFS over the NO ADT arm.” -Line217: „When comparing the No ADT to ST-ADT groups, no benefit was seen in terms of MFS with a HR of 0.89; CI: 0.69-1.14.” It is a kind of contradiction, please clarify the data and interpretation of the RADICALS-HD study.  

-It would be worthy to add one sentence about the possible effect of new technologies/treatment options in this clinical situation, like PSMA PET-CT, ARTA therapies etc.

Reviewer 3 Report

This study was reported the results of the McGill 0913 study. Generally, this paper is well written. The reviewer thinks that this paper has useful information for readers. However, the reviewer would like to suggest some critiques to make this paper as follows.

Major revision

1.     On line 24, the cause of death should be stated.

2.     ADT + bicalutamide is given as a long-term hormone therapy, please discuss the differences when compared to ADT alone.

3.     The small number of enrolled patients may be a vital weakness of this study. Therefore, we should avoid drawing definitive conclusions about the validity of this study.

Round 2

Reviewer 3 Report

none.